# Ouabain-Induced Changes in the Expression of Voltage-Gated Potassium Channels in Epithelial Cells Depend on Cell–Cell Contacts

**DOI:** 10.3390/ijms232113257

**Published:** 2022-10-31

**Authors:** Marcelino Cereijido, Lidia Jimenez, Lorena Hinojosa, Aida Castillo, Jacqueline Martínez-Rendon, Arturo Ponce

**Affiliations:** 1Department of Physiology, Biophysics and Neurosciences, CINVESTAV-IPN, CDMX 07360, Mexico; 2Molecular Medicine Laboratory, Unidad Academica de Medicina Humana y C.S, Campus UAZ Siglo XXI-L1, Universidad Autónoma de Zacatecas, Zacatecas 98160, Mexico

**Keywords:** ouabain, epithelial cells, potassium channels, patch clamp, Na^+^/K^+^-ATPase

## Abstract

Ouabain is a cardiac glycoside, initially isolated from plants, and currently thought to be a hormone since some mammals synthesize it endogenously. It has been shown that in epithelial cells, it induces changes in properties and components related to apical–basolateral polarity and cell–cell contacts. In this work, we used a whole-cell patch clamp to test whether ouabain affects the properties of the voltage-gated potassium currents (Ik) of epithelial cells (MDCK). We found that: (1) in cells arranged as mature monolayers, ouabain induced changes in the properties of Ik; (2) it also accelerated the recovery of Ik in cells previously trypsinized and re-seeded at confluence; (3) in cell–cell contact-lacking cells, ouabain did not produce a significant change; (4) Na^+^/K^+^ ATPase might be the receptor that mediates the effect of ouabain on Ik; (5) the ouabain-induced changes in Ik required the synthesis of new nucleotides and proteins, as well as Golgi processing and exocytosis, as evidenced by treatment with drugs inhibiting those processes; and (5) the signaling cascade included the participation of cSrC, PI3K, Erk1/2, NF-κB and β-catenin. These results reveal a new role for ouabain as a modulator of the expression of voltage-gated potassium channels, which require cells to be in contact with themselves.

## 1. Introduction

Ouabain is a cardiac glycoside, formerly isolated from plant sources [1], that, depending on its concentration, produces a wide variety of toxic or therapeutic effects [2,3], among which the best known is its positive inotropic action on the heart [4,5]. Ouabain binds to Na^+^/K^+^-ATPase, inhibiting its pumping action, which entirely accounts for its positive inotropic effect on heart [6,7]. For this reason, ouabain has been prescribed to treat congestive heart failure and supraventricular arrhythmias. Nonetheless, because this substance has a very narrow therapeutic range, it has been gradually discontinued and forgotten by physicians [8].

However, two facts have prompted a renewed interest in knowing more about the effects produced by ouabain: (1) it has been described that ouabain, like other cardiotonic steroids, is synthesized endogenously in some mammal species, including humans [9,10,11], which has led to the consideration of this compound as a hormone [12,13], whose physiological role is not entirely known; and (2) several studies have shown that Na^+^/K^+^-ATPase, in addition to functioning as an ion transport pump, also works as a receptor, which, after binding with ouabain and other cardiac glycosides, activates a variety of responses in cells through several signaling pathways [14,15,16].

In previous work, we focused on studying the effect that ouabain produces in epithelial cells, using mainly as a model the MDCK cells, a cell line derived from dog kidneys that is widely used for this purpose [17,18,19,20]. Thus, we have shown that ouabain influences several processes and components related to cell-to-cell contact and communication: (1) it enhances the intercellular contact produced by the tight junctions, which is demonstrated by an increase in transepithelial electrical resistance (TEER), a decrease in the transcellular flow of dextran concomitantly with an increase in the expression of claudins 1, 2 and 4 [21]; (2) ouabain also produces changes in adherens junctions, causing the enhanced expression of E-cadherin, β-catenin and γ-catenin [22]; (3) ouabain enhances gap junctional communication between cells arranged in mature epithelial monolayers [23,24] by modulating the membrane distribution of Cx32 and Cx43 [25]; and (4) ouabain speeds up the reestablishment of the epithelial phenotype in cells seeded at confluency after trypsinization, as evidenced by the expression of an apical cilium in MDCK cells [26].

Given that ion channels are fundamental components of all cells, both in excitable and non-excitable cells [27,28], in this work, we focused on analyzing whether ouabain induces changes in the expression of endogenous voltage-gated potassium channels of MDCK cells. 

Previous studies have described the electrical as well as the functional properties of the endogenous channels of MDCK cells [29,30,31]. They have been described as mainly expressing potassium and chloride channels [32,33,34,35] which are polarized; that is, they are expressed either in the apical or basolateral domains of the cells when they are arranged as mature epithelial monolayers [36,37]. It has also been described that when MDCK cells are trypsinized, they lose membrane and potassium channels, but they are able to restitute this shortly after seeding at confluence, in a process that requires the synthesis of new channels [38]. It has furthermore been found that the properties and magnitude of potassium currents in MDCK cells depend on cell-to-cell contacts because in the subconfluent state, the characteristics of potassium currents are distinct from those expressed in cells in mature monolayers [39,40].

Given these facts, in this work, we resorted to electrophysiological procedures (whole-cell patch clamping) to test, through the recording, analysis and comparison of the properties of ion currents, whether ouabain, at a nanomolar concentration (10 nM), induces changes in the expression of voltage-gated potassium channels in MDCK epithelial cells. 

## 2. Results

### 2.1. Biophysical Properties of the Endogenous Potassium Currents of MDCK Cells in Mature Epithelial Monolayers

To start, we recorded ion currents from cells assembled as mature epithelial monolayers (MEMs) which had been seeded at confluency three days before. To exclude anion currents, Cl^−^ was substituted by gluconate^−^ in both extracellular and intracellular solutions (see Methods). To generate these currents, we applied a stimulation protocol consisting of a series of voltage steps that, starting from a holding voltage (Vh) of −75 millivolts, switched to test voltages ranging from −80 to +80 mV in steps of +20 mV. Figure 1a shows a representative series of the traces of ion currents obtained under these conditions. These currents activated, increasing their magnitude over time, until they reached a steady-state level (Iss) which was maintained while the test pulse lasted. Both the promptness of activation and Iss increased with voltage. The magnitude of these currents depended, in part, on the size of the cells; for this reason we estimated the membrane surface by means of capacitance measurements (as described in the Methods) to describe the results rather than as current density (δI). Thus, Figure 1b shows the relationship between the average value (±SE) of δI*_ss,v_* (the current density at the steady-state for a given voltage) and the test voltage (*v*). 

In turn, for each of the current density values, the conductance density (δg*_ss,vt_*) was calculated by the chord conductance equation, derived from Ohm’s law:$${\delta g}_{ss,vt}=\frac{{\delta I}_{ss,vt}}{\left(v_{t}-v_{inv}\right)}$$
where *v_t_* is the test voltage and *v_inv_* is the inversion potential, a constant value (−75 mV) that was obtained from tail-current assays.

As shown in Figure 1c, the relationship between *δgss* and *vt* has a sigmoidal profile. 

So it was fitted to the following equation (Hill’s equation):$$\delta gss\left(v\right)=\frac{\delta G_{max}}{1+e^{-\left(\frac{v-v_{50}}{Hs}\right)}}$$
where $\delta G_{max}$ is the maximum conductance density, $v_{50}$ is the voltage value at which half of the response is obtained and *Hs* is the Hill’s slope.

On the other hand, the onset phases of the traces of current, at any given test voltage, were fitted to the following equation to obtain the value of the time constant of activation (*τ_on,v_*):$$I_{t,v}=Iss_{v}\left(1-e^{-\frac{t}{\tau_{on,v}}}\right)$$

Figure 1d shows the relationship between the values of *τ_on_* (±SE) and the voltage in the range from 0 to +80 mV. In turn, these data were fitted with the following exponential decay function:$$\tau_{on,v}=\tau_{on,0}\;\cdot e^{-\frac{v}{\tau_{v}}}$$
to obtain the parameters, $\tau_{on,0}$ (the value of $\tau_{on,v}$ when *v* = 0) and *τ_v_* (the constant of decay of $\tau_{on,v}$ over voltage). In this way, we analyzed all the series of currents obtained, each obtained from a different cell, to find the average (±SE) value of each parameter (*δG_max_*, *V*_50_, *Hs*, *τ*_0_ and *τ_v_*). Table 1 summarizes the results obtained after recording a sample of 32 individual cells in MEMs.

### 2.2. Ouabain Induced Statistically Significant Changes in the Biophysical Properties of Potassium Currents in Cells Assembled in Mature Epithelial Monolayers (MEMs)

Next, following the procedure described above, we tested whether ouabain would induce changes in the properties of potassium currents in cells assembled as MEMs. For this purpose, we grew cells as MEMs on glass coverslips. After three days of maturity, we grouped the coverslips into pairs, one of which we treated with 10 nM of ouabain while the other, we did not (control). Thus, we recorded series of currents of a given number of cells of each pair for different treatment times (1, 6, 12 and 24 h). Furthermore, for each group, we calculated the average values of the different parameters described in the previous section, to make a statistical comparison between the groups that were and were not treated with ouabain at each treatment time.

As shown in Figure 2, ouabain significantly increased the magnitude of potassium currents: panel A shows a representative series of currents obtained at the different times with and without treatment with ouabain, as well as before treatment, while panels B and C compare the biophysical parameters of potassium currents.

The *δG_max_* was significantly higher after 6 h of treatment with ouabain (Figure 2(B11)), reaching a maximum value after 12 h of treatment; then, it had a slight decline at 24 h of treatment, however, it remained significantly higher than the corresponding value of the control groups. The V50 also changed significantly, but this was only up to 12 h of treatment (Figure 2(B12)). The Hill’s slope value did not have a statistically significant change in any of the treatment times (Figure 2(B13)).

Treatment with ouabain also affected the activation kinetics of potassium currents (Figure 2(C1)): the average value of *τ*_0_ increased significantly from 6 h of treatment (Figure 2(C11)), while *τ_v_* did not change significantly at any treatment times (Figure 2(C12)).

These results therefore indicate that ouabain effectively induced changes in the biophysical properties of potassium currents, suggesting that it promotes the expression of potassium channels in MDCK cells when they are assembled as MEMs. 

### 2.3. Ouabain Accelerated the Recovery of Voltage-Gated K^+^ Channels Lost during Trypsinization When Cells Were Seeded at Confluency

As mentioned before, we previously described that trypsinization causes cells to lose membrane as well as an important amount of potassium channels, as the density of ion currents decreases significantly; however, cells eventually restitute their former condition by synthetizing new channels [39,40]. On the other hand, we also described evidence suggesting that ouabain influences cells by accelerating the maturation process of an epithelial monolayer, as demonstrated in results in which it accelerated ciliogenesis in cells seeded at confluence [26]. In view of these two antecedents, we sought to test whether ouabain would accelerate the recovery of potassium currents after trypsinization.

For this purpose, we trypsinized MDCK cells that had been grown and maintained as MEMs for three days, and re-seeded them at confluency, putting 10 nM of ouabain in the culture medium immediately after re-seeding. Then, we recorded K^+^ currents from individual cells, at several times after plating (20 min and 1, 3, 6, 12, 24 and 48 h), and compared their biophysical parameters with those of cells not treated with ouabain at the equivalent times. 

Figure 3 shows, in panel A, several series of potassium currents that were recorded from cells before trypsinization (*in blue*) and at different times after being trypsinized and seeded at confluence, both under control conditions (*in black*), and with 10 nM of ouabain in the extracellular medium (*in red*), which was added from the moment of seeding. Panel B shows several graphs (B1–B6) that compare the average values (±SE) of the different parameters (membrane area, *δG_max_*, *V*_50_, *Hs*, *τ*_0_ and *τ_v_*) between the control groups and the cells treated with ouabain at different times after seeding. In all these graphs, the black circles indicate the control results, and the red ones, those obtained from cells treated with ouabain. Moreover, the horizontal lines (*in blue*) correspond to the average (±SE) values obtained from the cells, before trypsinization, for each given parameter.

As shown in graph B1, the membrane area was significantly reduced after trypsinization, but gradually recovered until it reached its initial value after 12 h; ouabain did not influence this process nor did it influence the speed of activation of currents. As shown in graphs B2 and B3, the values of *τ*_0_ and *τ_v_* were not statistically different between the control groups and those treated with ouabain, or in the case of *V*_50_ and *Hs*.

However, trypsinization did significantly affect the maximum conductance density (*δG_max_*), as shown in Figure 3(B4). In both, untreated and ouabain-treated cells, the *δG_max_* was significantly reduced compared to the average value of cells before trypsinization (*blue asterisks*); nonetheless, this parameter gradually recovered such that after a given time, it was no longer statistically different from the value it had before trypsinization (*blue horizontal lines*). In the case of cells not treated with ouabain, the *δG_max_* ceased to be significantly lower than that of cells before trypsinization, until 24 h after seeding, while in cells treated with ouabain, this time was reduced to 12 h. 

Considering the *δG_max_* is an estimate of the density of K^+^ channels in the membrane, these results might suggest that ouabain favors a faster recovery of K^+^ channels that were lost in the process of trypsinization.

### 2.4. Ouabain Did Not Produce Changes to Potassium Currents in Subconfluent Cells

Next, we tested how ouabain acted in subconfluent cells. For this purpose, we trypsinized and seeded cells on glass coverslips at a low density (subcultivation ratio, 1:15) and kept them in culture for two days; subsequently, we treated some of these coverslips with 10 nM of ouabain or not (control) and kept them for 6 and 12 h, after which we made recordings of the potassium currents from cells of each group, as already described. After analyzing the K^+^ currents, we obtained the mean (±SE) values of the different parameters (*δG_max_*, *V*_50_, *Hs*, *τ*_0_ and *τ_v_*) and made statistical comparisons between the untreated and treated groups.

Figure 4 shows in panel A, the series of potassium currents representative of the different conditions. The series (*in blue*) on the left side, again, was obtained from a cell in MEMs before trypsinization. The next series (control, 0 h) was obtained from a cell 2 days after having been seeded at low density but before adding the ouabain to the extracellular medium. The other series correspond to what was obtained at the times of 6 and 12 h with or without treatment with ouabain. In panel B are shown different graphs (B1–B6) that compare the average values (SE) of each parameter, with and without ouabain (*red and black circles*), at 6 and 12 h of treatment, as well as those from cells in MEMs (*horizontal blue lines*).

As shown in graph B1, the *δG_max_* was lower than that obtained from the cells in the MEMs (blue horizontal lines); however, there was no significant difference between the groups treated and not treated with ouabain at any of the treatment times. Moreover, the *τ*_0_ of subconfluent cells (graph B2) was noticeably higher than that of cells in MEMs without there being a significant difference between treated and untreated groups at any time. The membrane surface (graph B4) is also noticeably smaller in subconfluent cells than in cells in MEMs, but there were no significant differences between the control and ouabain-treated groups at any time. For the other parameters (*V*_50_, *Hs*, and *τ_v_*) there was no difference, neither between groups at different times, nor in comparison with those of cells in MEMs.

These results suggest that for ouabain to effectively induce a change in the magnitude and biophysical properties of potassium currents, cells are required to be touching each other.

### 2.5. Removal of Extracellular Ca^2+^ Abolished the Changes Induced by Ouabain on Potassium Currents in Cells in MEMs

The previous results prompted us to test the effect of ouabain on potassium currents, again in cells in MEMs, but this time without calcium in the culture medium, since it has been shown that lowering calcium from the extracellular medium disrupts cell-to-cell contacts [41,42]. To this end, cells were seeded at confluency in glass coverslips and maintained for three days; after that, the coverslips were divided into two groups. The first group was kept in a normal culture medium, while the medium of the second group was incubated with a low calcium medium for 1 h before treatment with ouabain (6 h, 10 nM); then, potassium currents were recorded from either group, with or without calcium and with or without ouabain. After analysis, the average values of the same parameters as in the previous tests (*δG_max_*, *V*_50_, *Hs*, *τ*_0_ and *τ_v_*) were compared to determine if there was a statistically significant difference between the groups in the presence or absence of calcium.

Figure 5 shows, in panel A, two pairs of series of currents: the left side corresponds to the control and the group treated with ouabain in the presence of normal extracellular calcium (NC), while the right side corresponds to the control and the group treated with ouabain, but in monolayers with low calcium (LC). Panel B shows graphs that compare the relationships between the current density, conductance density, g/gmax and *τ_on_*, and the test voltage. Panel C shows histograms and the result of statistical comparisons between the control groups and those treated with ouabain, either in normal (NC) or low calcium (LC). As shown therein, the results indicated that none of the biophysical parameters of the potassium currents were modified by ouabain in the absence of calcium, while in its presence, all the parameters underwent changes, as we already described above. 

### 2.6. Na^+^/K^+^-ATPase Is the Receptor Mediating the Action of Ouabain on Potassium Currents

As mentioned before, Na^+^/K^+^-ATPase, in addition to functioning as an ionic pump, works as a receptor, so it has been shown to be the receptor by which ouabain stimulates a variety of physiological processes. We have already shown that in MDCK cells, Na^+^/K^+^-ATPase mediates ouabain, as well as digoxin and marinobufagenin to modulate GJIC and TER [21,22,23,24,25,43]. Therefore, in this work, we sought to know whether Na^+^/K^+^-ATPase also mediates the effect that ouabain produces on potassium currents. For this purpose, we assayed the effect of ouabain on potassium currents in MDCK-R, a variant of MDCK, which was rendered unsensitive to ouabain by chemical mutagenesis, induced by ethyl methanesulfonate [44].

Thus, we compared the effect of ouabain on potassium currents on both wild (MDCK-W) and resistant (MDCK-R) cells. We seeded cells at confluence in coverslips and, after 3 days, we incubated them with ouabain for 12 h, after which we made whole cell recordings, both in cell monolayers which had been treated with ouabain and with control monolayers. Then, we analyzed the currents to obtain the average values of the different parameters and tested if there was a statistically significant difference between them. 

Figure 6 shows, in panel A, a representative series obtained under the same experimental conditions, of MDCKW and MDCKR cells. Panel B shows in graphs (B1–B4) the relationship between voltage and current density, conductance density, g/gmax and tauon, while panel C shows histograms that compare the different parameters, both in MDCK-W cells and in MDCK-R. As indicated there, in MDCKW cells, ouabain produced statistically significant changes in all parameters, but not in MDCK-R cells. 

These results suggest that the Na^+^/K^+^-ATPase is the receptor that mediates the effect of ouabain on potassium currents in MDCK cells.

### 2.7. Ouabain Stimulated the Synthesis, Assembly and Exportation of New Ion Channel Units

To know if the changes in the characteristics of potassium currents induced by ouabain required the synthesis of new protein units (channels), we tested the effect, of a set of inhibitors and blockers at different stages of the production of new units. These included: (1) inhibitors of RNA synthesis, actinomycin D [45,46,47] and α-amanitin [48,49]; (2) inhibitors of protein synthesis, cycloheximide [50,51,52] and puromycin [53,54,55]; (3) inhibitors of Golgi processing, brefeldin A [56,57,58] and tunicamycin, which prevents N-linked glycosylation of new proteins [59,60,61,62]; and (4) inhibitors of exocytosis, cytochalasin B [63,64] and Exo1 [65].

To test the effect of these inhibitors, we separately made recordings of potassium currents from cells in mature monolayers under four conditions corresponding to the combinations of with or without ouabain (during 6 h of treatment), and with or without the given inhibitor (the dose used for each inhibitor is shown in Table 2 of Methods). From these data, we calculated the average value of the density of conductance for each group and compared the corresponding values of the ouabain groups without the given inhibitor versus the ouabain groups with it, to determine if this given inhibitor blocked the effect of ouabain. 

Figure 7 shows the average value (±SE) of conductance density, for each of the four different combinations (of ouabain and inhibitor), as well as the result of the statistical comparison between the groups treated with ouabain, for the different inhibitors described above. As shown there, actinomycin significantly inhibited the increase in conductance density induced by ouabain, while CX5461 did not produce a statistical inhibition of the response. The two blockers of protein synthesis, cycloheximide and puromycin, produced a very significant reduction in the response induced by ouabain, which strongly suggests that the synthesis of new protein units is required for this response to occur. This is further reinforced by the result with brefeldin A, which also produced a significant reduction in the stimulus produced by ouabain. On the other hand, both exocytosis inhibitors produced a significant reduction in the effect induced by ouabain, indicating that this cellular process is required for the incorporation of new molecular components.

These results suggest, therefore, that ouabain stimulates the production of new ion channels, through the synthesis of new copies of mRNA and proteins, as well as processing in the Golgi apparatus and finally, exocytosis to be incorporated into the cell membrane.

### 2.8. Signal Transduction Pathways Involved in the Action of Ouabain on Potassium Currents

Using an experimental strategy such as the one described in the case of synthesis inhibitors, we also tested the participation of various components of signaling pathways through evaluating the action of ouabain on *δG_max_*, in the absence or presence of several specific inhibitors of components of signaling pathways (Figure 8).

To probe the involvement of c-Src, we tested the effect of PP2, a potent, reversible and selective inhibitor of the Src family of protein tyrosine kinases [66]. As Figure 8a shows, PP2 significantly (*p* < 0.001) abolished the ouabain-induced enhancement of *δG_max_*, indicating that cSrc effectively participates in the signaling cascade.

Another mechanism of signal transduction, independent of SrcK, that Na^+^/K^+^-ATPase has been shown to be involved in, is the activation of PI3K [67,68]. Therefore, we tested Wortmannin, a potent inhibitor of PI3K enzymes [69,70]. This compound also significantly (*p* < 0.05) abolished the effect of ouabain, indicating that PI3K participates in the signaling cascade.

Next, to probe the involvement or the Raf/MEK/ERK pathway, we tested the effect of two inhibitors: PD 184161, a potent and selective inhibitor of the phosphorylation of ERK1/2 [71], and FR180204, a potent, cell-permeable inhibitor of ERK1 and ERK2 [72]. As shown in Figure 8c,d, both inhibitors significantly (*p* < 0.005) abolished the response induced by ouabain, indicating the participation of Erk1/2.

On the other hand, it has been described that ouabain induces the activation of the nuclear factor-κ B (NF-κB), which in turn plays a key role in the regulation of the BDNF and WNT-β-catenin signaling cascades [73,74]. The fact that IKK-16, a potent inhibitor of IκB kinases [75,76,77] significantly abolished the ouabain-induced enhancement of *δG_max_*, indicates that NF-κB participates in the signaling cascade (Figure 8e).

We also tested if β-catenin plays a role by probing the effect of FH535, an inhibitor of β-catenin-mediated transcription [78,79]. As Figure 8f shows, this caused a significant (*p* < 0.05) decrease in the response induced by ouabain, indicating the role of β-catenin in the signaling cascade.

Finally, we tested Y-27632, a potent inhibitor of ROCKs [80,81], to determine whether the Rho/ROCK was also involved in the ouabain-induced change in potassium currents. However, as Figure 8g shows, this treatment failed to abolish the effect caused by ouabain, excluding the Rho/ROCK pathway from the signaling cascade.

## 3. Discussion

It is widely known that the toxic and therapeutic properties of ouabain are related to changes in the ion transport mechanisms of cells of different origins, among which cardiac and smooth muscle cells stand out [82,83,84]. Thus, it has been widely described that, at doses in the micromolar range, ouabain inhibits the pumping capacity of NAK, causing an imbalance in sodium and potassium homeostasis, which in turn promotes the activation of various ion transport mechanisms, including ion channels and transporters for sodium, potassium and calcium. This effect, known as positively inotropic, has been the explanation for ouabain’s ability to increase the contraction force in the heart and smooth muscle [85,86,87]. In epithelia, it has also been described that ouabain influences potassium permeability. At concentrations above its hormonal range, it inhibits the sodium–potassium pump, which causes an accumulation in intracellular sodium. This in turn slows down the calcium–sodium exchanger, resulting in an intracellular calcium accumulation and consequently, an increase in potassium permeability due to calcium-dependent potassium channels [88]. However, the possibility that ouabain, as a hormone, (i.e., at doses in the nanomolar range) induces changes in the properties of ion transport mechanisms, including ion channels, had not been evaluated so far, even though evidence had already been described that endogenous ouabain induces modifications in physiological or pathological processes directly related to ion flows, such as hypertension and electrolyte reabsorption in the kidney [89,90,91].

Thus, in this work, we evaluated the possibility that ouabain, at a dose in the hormonal range (10 nM), influences the expression of potassium channels in epithelial cells (MDCK) using electrophysiological methods (whole-cell patch clamp) and biophysical criteria. We found that ouabain, at a concentration in the nanomolar range, induced significant changes in the biophysical properties of I_K_ when the cells were assembled as mature epithelial monolayers (MEMs). Given that the biophysical properties of ion currents reflect the amount, as well as variety of ion channels expressed in the membrane, the results obtained indicate that ouabain, besides enhancing the density of conductance to potassium (dG_max_), also induces changes in other biophysical parameters (*V*_50_, Hill’s slope and *τ_on_*). This suggests that ouabain not only promotes the expression of more channels in the membrane, but also induces the expression of new types of channels, different from those being expressed otherwise. This hypothesis is further supported by the fact that inhibitors of the synthesis, processing and export of new proteins were able to reverse the effect caused by ouabain on I_K_´s properties.

On the other hand, as interesting as the fact that ouabain influences I_K_, so is the finding that it is necessary that the cells touch each other. As described above, the result that ouabain did not induce changes in the I_K_ in subconfluent cells, made us guess that it is required that cells be touching. This hypothesis is supported by the fact that disrupting the cell-to-cell contacts of cells in mature monolayers, by lowering the extracellular calcium, abolished the effect of ouabain on I_K_, and even further, by the fact that FH535, an inhibitor of β-catenin/Tcf-mediated transcription [78,79], partially, yet significantly, abolished the response induced by ouabain on I_K._ In relation to this, we described that in cells in mature monolayers, ouabain induced a greater expression of β-catenin, and promoted its translocation to the nucleus [22].

Apart from the result obtained for FH535, which suggested the participation of a WNT- β-catenin pathway, the results with kinases inhibitors indicated the involvement of PI3K/AKT and cSRC/MEK/Erk1/2 pathways. This raises the possibility that ouabain can simultaneously trigger several responses, separately promoting the expression of distinct types of K^+^ channels. However, this hypothesis cannot be tested further because, so far, there is a very limited description of the variety of endogenous ion channel types of MDCK cells, and all of them are based on electrophysiological studies [31,92]. Nonetheless, the biophysical properties of MDCK´s I_K_ suggest the expression of voltage-gated potassium channels (Kvs), an important family of the 6TM K channels, composed of 12 subfamilies (Kv1 to Kv12), each of which, in turn, has several molecular variants, each produced by a distinct gene. These channels are assembled as tetramers, and can be constituted by identical (homomeric) or different subunits (heteromeric), as long as the different subunits belong to the same subfamily (e.g., Kv1.1 with Kv1.2; Kv7.2 with Kv7.3) [93]. Among the distinct subfamilies of Kvs, those that best match the characteristics of MDCKs cells’ I_K_ (delayed rectification, v50 values above +20 mV and slow activation) could be Kv2.x. Kv1.5 channels could also match this biophysical description since they also produce delayed rectifying currents. Yet, further work with molecular and immunohistochemical approaches is necessary to resolve the molecular identity of the types of K^+^ channels that account for the endogenous potassium currents of MDCKs and those that ouabain influences.

Regarding the physiological and/or pathological relevance of these findings, it is worth wondering what role is played by those channels whose expression is modified by ouabain, as well as knowing if the effect of ouabain described here applies only to these cells or is a phenomenon that is observed in all epithelial tissues or even in all types of cells. The fact that this action depends on the existence of contacts between cells suggests that it could be limited to epithelial cells. Given that MDCK cells are thought to be derived from the distal tubule of the kidney [94,95,96], whose function is the secretion of potassium [97,98,99], it is possible that the channels that modulate ouabain are involved in potassium secretion. It is generally accepted that the renal outer medullary potassium channel (ROMK), an inwardly rectifying potassium channel, is the K^+^ secretory channel in the mammalian distal nephron [100]. However, MDCK showed no inwardly rectifying currents, so it is unlikely that inwardly rectifying potassium channels are those stimulated by ouabain.

Nonetheless, recent in vitro and in vivo studies have provided evidence that large-conductance Ca^2+^-activated K^+^ channels (BK^+^ or maxi K^+^) also secrete K^+^ in renal tubules [101,102,103]. As it has been reported that MDCK cells express calcium-dependent potassium channels [104], it is possible that calcium-dependent K^+^ channels are among the types of channels that contribute to MDCK cells’ I_K_ currents which ouabain modulates.

Another role that has been attributed to potassium channels has to do with the regulation of the cell cycle. There is substantial evidence that several K^+^ channels play a role in the cell cycle and proliferation by means of both permeation-related and unrelated mechanisms [105]. Several members of the Kv family are related to the cell cycle: Kv1.3, Kv1.5, Kv11.1 (hERG) and Kv10.1 (Eag1). For this reason, they are also associated to pathologies related to the regulation of the cell cycle, among which cancer stands out [106,107].

On the other hand, it has been suggested that ouabain and other cardiac glycosides are related in one way or another to cancer [108,109,110,111], although the complex mechanisms and signaling pathways have not yet been identified, nor all the components involved; as described in this work as well, it is likely that some types of potassium channels are involved. 

## 4. Materials and Methods

### 4.1. Cell Culture

Starter MDCK-II cells, referred to here as MDCK-W, were obtained from the American Type Culture Collection (MDCK, CCL-34). MDCK-R, a subclone highly resistant to ouabain, was kindly provided by Dr. Louvard (Pasteur Institute). 

For production and maintenance, cells (both MDCK-W and MDCK-R) were grown in a 5% CO_2_ atmosphere at 36.5 °C, in Dulbecco’s Modified Eagle Medium (DMEM; Gibco, Waltham, MA, USA), supplemented with 10,000 U/μg/mL of penicillin-streptomycin (Cat. 15140122, Thermo Fisher Scientific, Waltham, MA, USA) and 10% fetal bovine serum (Gibco). 

For the electrophysiological assays, the cells in mature epithelial monolayers (MEMs) were trypsinized and seeded at 80% confluence on glass coverslips, placed in 24-well cell culture plates, with CDMEM + 10% FBS, and kept for 3 days after seeding.

### 4.2. Electrophysiological Recording of Cells

Potassium currents were recorded using the whole-cell patch clamp technique following standard procedures, as described elsewhere [112,113]. Briefly, micropipettes with a tip resistance of 2–5 MΩ were made from borosilicate glass tubing (cat. 34500-99, Kimble Chase, Vineland, NJ, USA), with a horizontal puller device (P-87, Sutter Instrument Co., Novato, CA, USA), backfilled with intracellular solution (see Solutions) and attached, through a pipette holder, to a piezoelectric-driven micromanipulator (PCS6000, Burleigh Co., Newton, NJ, USA). Then, pipettes were driven to cells to make gigaseals, then whole-cell recordings of ion currents, monitoring the process with an inverted microscope (Diaphot 300, Nikon, Tokyo, Japan). 

Glass coverslips containing MDCK cells were immersed in a chamber filled with extracellular solution (see Solutions) and continuously perfused. Glass tubing filled with 2% agarose in 500 mmol/L of KCl was set to make electrical contact between the bathing solution and the reference electrode, which was immersed in 500 mM of KCl. Patch rupture was achieved by suction after the gigaseal reached values greater than 2 GΩ (typically 5 GΩ).

The stimulation protocol consisted of a series of voltage square pulses, from which a holding potential (Vh) of −75 mV changed to a test potential from −80 mV to +80 mV in steps of 20 mV, lasting 1 sec each, then switched back to Vh. A p/4 protocol was set to subtract linear components. Voltage pulse protocols and recordings of ion currents were made with a patch clamp amplifier (8900, DAGAN Corp, Minneapolis, MN, USA), controlled by a dedicated software suite (pClamp 8.0, Axon Instruments Inc., Grand Terrace, CA, USA).

### 4.3. Measurement of Membrane Capacitance

A capacitive current transient was induced by a hyperpolarizing square pulse of voltage, from −100 to −110 mV, and recorded at 10 KHz. Membrane capacitance was calculated offline by integrating the area of the capacitive transient at the onset of the pulse, then dividing the integrate by the amplitude of the pulse (−10 mV), according to the following equation:$$Cm=\frac{\int_{t0}^{\infty }Ic\cdot \Delta t}{\Delta V}$$
where cm is the membrane capacitance, *Ic* the capacitive current and Δ*V* the amplitude of the voltage pulse (−10 mV). Calculation of the integrate was made with the Clampfit module of pClamp 8.0 (Molecular Devices, Grand Terrace, CA, USA).

### 4.4. Solutions

The pipette (intracellular) solution was composed of (mmol/L): 135 K-gluconate, 5 Na-gluconate, 1 MgCl_2_, 5 glucose, 10 HEPES, 10 EGTA, pH 7.4, adjusted with KOH. The extracellular solution composition consisted of (mmol/L): 140 Na-gluconate, 5 K-gluconate, 3 CaCl_2_, 1 MgCl_2_, 5 glucose, 10 HEPES, pH 7.4 adjusted with NaOH. 

### 4.5. Chemicals and Drugs

All salts, chemicals and drugs were purchased from Sigma-Aldrich (St. Louis, MO, USA) Table 2 shows the catalog number, as well as the stock concentration (in DMSO) and the dose used in this work.

### 4.6. Statistical Analyses

The data collected in this work was processed and analyzed statistically using the Microsoft Office 365 Excel application and Sigmaplot 14.5. The results shown are the product of three independent experimental trials. The number of data is indicated in the figures and in the text. The data are represented as the average value and dispersion as the standard error of the mean (SE). Statistical analysis, as indicated in the text and figures, consisted of simple paired comparisons made, either with a *t*-test or Mann–Whitney rank sum test. A minimum criterion of *p* < 0.05 was considered for a statistically significant difference.

## 5. Conclusions

The results of this work demonstrate that ouabain (10 nM) induced statistically significant changes in the biophysical properties of potassium currents of MDCK epithelial cells, which suggests that it promotes the expression of a greater number of potassium channels, which may be new types of channels, not expressed in the absence of ouabain induction. It also accelerated the recovery of K^+^ channels lost during trypsinization. However, the cells were required to be in contact with each other for this induction to be effective. They also suggested that Na^+^/K^+^-ATPase might be the receptor that mediates the action of ouabain, which in turn triggers a signaling cascade involving several pathways, including cSRC, Erk1/2, PI3K, NF-kβ and β-catenin.

## Figures and Tables

**Figure 1 ijms-23-13257-f001:**
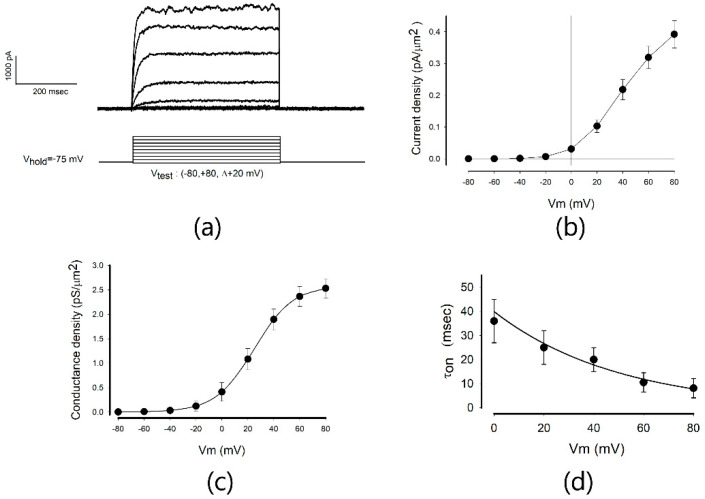
Properties of potassium currents in mature monolayer (MM) cells. (**a**) Representative series of current recordings, obtained in response to the series of square pulses of voltage represented below. Each trace is the response to a different test voltage. (**b**) The mean (±SE) current density versus the test voltage, in the range of −80 to +80 mV. The magnitude of the steady-state current was divided by the membrane area (which was estimated through capacitance measurements) to obtain the current density value. (**c**) The relationship between the membrane conductance density and the membrane voltage. Each point indicates the mean (±SE) value of conductance density obtained from 25 individual trials on cells conformed as mature monolayer (MM) cells. The solid line resulted from fitting these data to a sigmoidal function (Hill´s equation) to obtain the parametric values of Gmax, Hill´s slope and V50, as described in the text. (**d**) The relationship between the mean value (±SE) of the time constant of activation (ton) of ionic currents versus the test membrane voltage in the range of 20 to +80 mV. These values were obtained by fitting the rising phase of current traces to an exponential function. The solid line resulted after fitting the data to an exponential decay function, from which the parametric values of *τ*_0_ and *τ_v_* were estimated as described in the text. The mean values (±SE) were from 25 individual measurements of cells.

**Figure 2 ijms-23-13257-f002:**
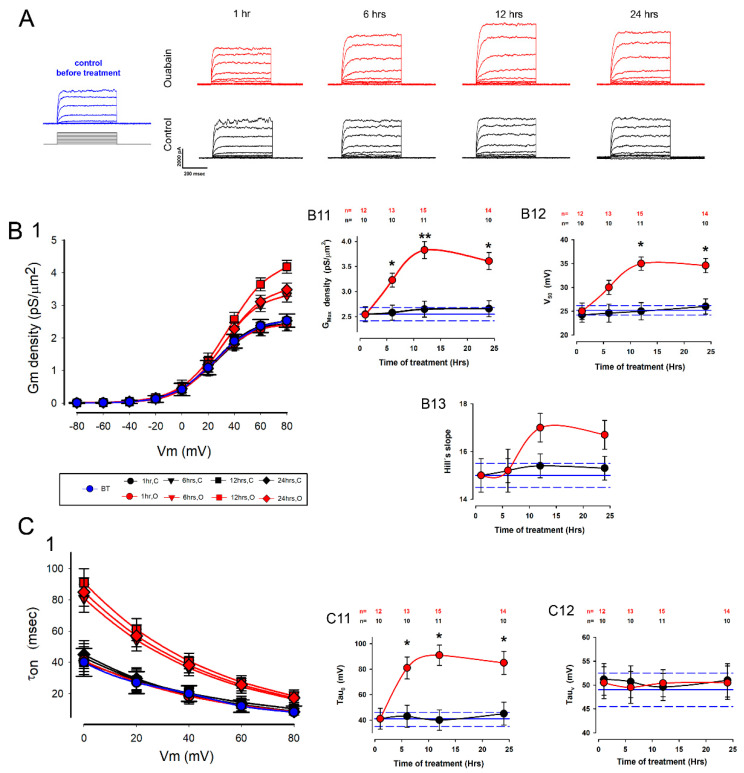
Effects of ouabain on the potassium currents of cells in mature monolayers (MM). (**A**) A representative series of currents recorded in response to the same voltage stimulation protocol, in cells under different treatment conditions. The series depicted in blue on the left side is representative of what was observed before treatment; the red colored series are representative of what was obtained at different treatment times with 10 nM of ouabain, while the black series are representative of the corresponding controls. (**B**) (**B1**) shows the relationships between the Gm density and the voltages for the different treatment times with ouabain (*in red*) and the controls (*in black*). The relationship between the recordings before treatment (in blue) is included. In the graph, the different plots are superimposed, as indicated in the box at the bottom; the circle, downward triangle, square and rhombus correspond to the times of 1, 6, 12 and 24 hours after treatment with ouabain. (**B11**–**B13**) compare the parametric values of the G_max_ density, V_50_ and Hill’s slope of treated (red) and untreated (black) cells at the different times of treatment. (**C**) (**C1**) shows the relationships between the activation time constant (tau on) and the voltage of the different treatment times with ouabain (in red), the controls (in black), and the relationship of the registers before treatment (in blue); in the graph, the different plots are superimposed, in the same way as in plot (**B1**). (**C11**,**C12**) compare the parametric values of *τ*_0_ and *τ_v_* of treated (red circles) and untreated (black circles) cells at different treatment times. In plots (**B11**–**B13**,**C11**,**C12**) the horizontal, blue lines indicate, as a reference, the mean (solid) and the error (dashed) of each parameter estimated from the cells before treatment. The asterisks denote a statistically significant difference between the means (±SE) of treated versus untreated cells; (*) indicates a *p* < 0.05 and (**) indicates *p* < 0.001.

**Figure 3 ijms-23-13257-f003:**
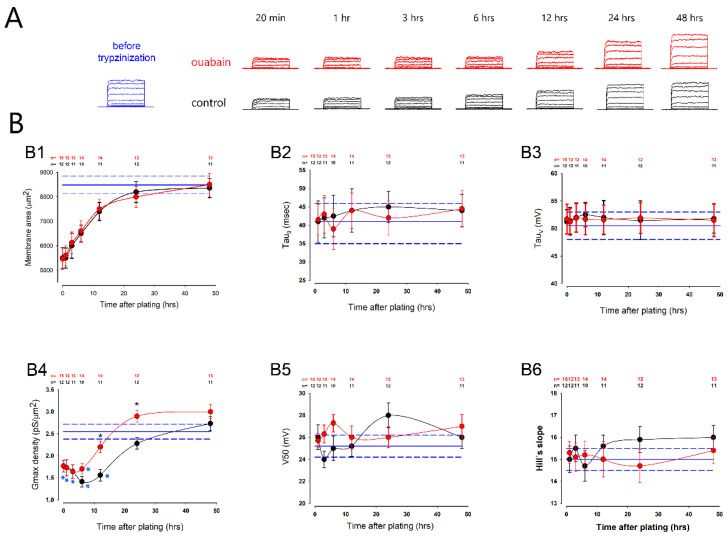
Ouabain accelerated the recovery of potassium currents after trypsinization. (**A**) The series of representative currents, recorded at different times after trypsinization and re-seeding at confluence. At the top, in red, are shown the series of currents recorded from the cells treated with ouabain. At the bottom, in black, are shown the series of currents without treatment with ouabain. The series shown on the left in blue corresponds to a series of currents recorded from a cell in a mature monolayer, before trypsinization. (**B**) Graphs (**B1**–**B6**) compare the average values (±SE) of the (**B1**) membrane surface, (**B2**) *τ*_0_, (**B3**) *τ**_v_*, (**B4**) density of gmax, (**B5**) V_50_ and (**B6**) Hill’s slope. In all these graphs, the red circles correspond to cells treated with ouabain and the black ones, to untreated cells. The blue lines show the corresponding values obtained from cells before trypsinization. Black asterisks indicate a statistically significant difference between the control and ouabain-treated groups for each time. Blue asterisks denote a statistically significant difference as compared to the mean value before trypsinization (*blue lines*). The numbers at the top of the graphs indicate the number of repetitions. (*) denotes *p* < 0.05.

**Figure 4 ijms-23-13257-f004:**
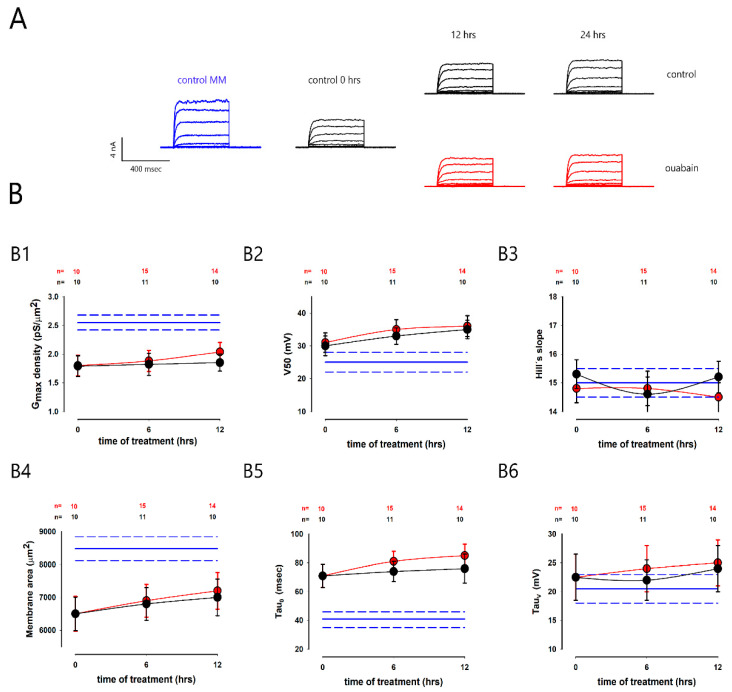
Effect of ouabain on subconfluent cells. (**A**) A representative example of series of currents obtained from individual cells under different conditions: the blue series was obtained from a cell in a mature monolayer, after three days of being seeded at confluence; the series marked as ‘control 0 h’ was recorded from a cell, two days after being seeded at subconfluence; and the other series shown were recorded from cells that were either untreated or treated with ouabain for 12 and 24 h. (**B**) Graphs (**B1**–**B6**) compare the mean values (±SE) of the: (**B1**) maximum conductance density, (**B2**) V_50_, (**B3**) Hill’s slope, (**B4**) membrane area (**B5**) *τ*_0_ and (**B6**) *τ**_v_*, both under control conditions (*black circles*) and treated with ouabain (*red circles*). In each graph, the horizontal blue lines indicate the corresponding values for mature monolayer (MM) cells. The asterisks show a statistically significant difference between the values represented by the circles and the corresponding values of cells in the mature monolayer (MM). The numbers at the top of each graph indicate the number of repetitions.

**Figure 5 ijms-23-13257-f005:**
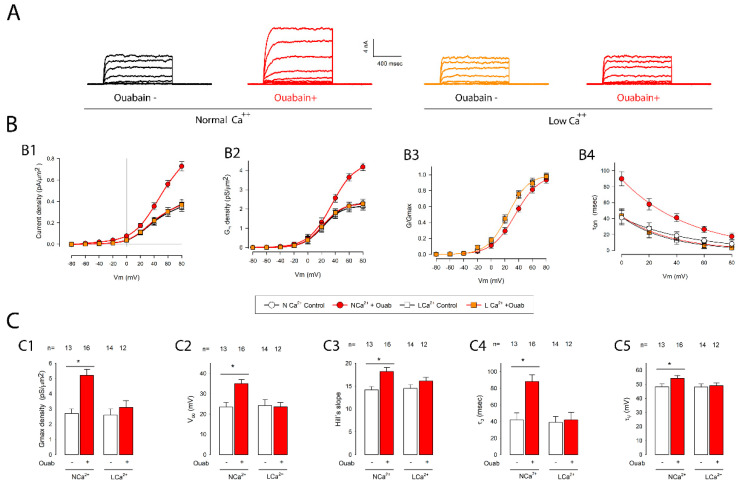
In the absence of extracellular calcium, ouabain had no effect on the potassium currents of cells in MEMs. (**A**) Series of currents representative of the different experimental conditions. The pair of series on the left side were obtained from cells in a mature monolayer, which had been either untreated or treated with ouabain for 6 hours. The pair of series on the right side was obtained from cells in mature monolayers from which calcium had been removed from the extracellular medium before treatment with ouabain. (**B**) The graphs (**B1**–**B4**) show the relationship between the different parameters (current density, conductance density, g/gmax and tau_on_) and the transmembrane voltage for the different experimental groups (with and without calcium, and with and without ouabain). The symbols for each group are shown in the box located below (**B2**,**B3**). (**C**) The bar charts (**C1**–**C5**) show the average values (±SE) obtained in each experimental group as well as the result of the statistical comparison between them. The numbers above the bars indicate the number of repetitions in each group. The asterisks above the horizontal lines denote a statistically significant difference between the means (±SE) of treated versus untreated cells; (*) indicates a *p* < 0.05.

**Figure 6 ijms-23-13257-f006:**
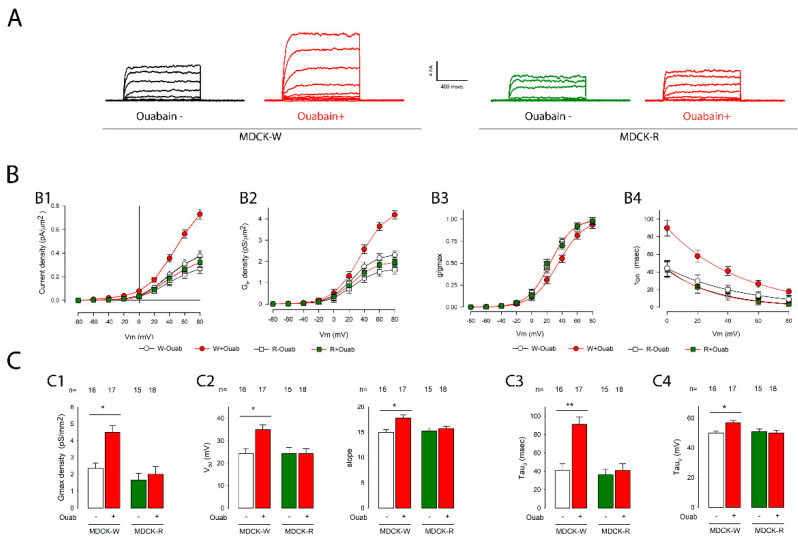
Na^+^/K^+^-ATPase is the receptor that mediates the effect of ouabain on potassium currents. (**A**) Series of recorded representative streams of mature monolayer cells, of both wild (MDCKW, left) and resistant cells (MDCKR, right), both untreated and treated with 10 nM of ouabain for 6 hours. (**B**) Graphs (**B1**–**B4**) show the relationship between the different variables (current density, conductance density, g/gmax and tauon) and the membrane voltage for the different groups (MDCKW and MDCKR both without and with treatment with ouabain). The symbols of each group are indicated at the bottom of the graphs. (**C**) Bar charts (**C1**–**C4**) show the average values (±SE) obtained in each experimental group as well as the result of the statistical comparison between them. The numbers above the bars indicate the number of repetitions in each group. The asterisks above the horizontal lines denote a statistically significant difference between the means (±SE) of treated versus untreated cells; (*) indicates a *p* < 0.05 and (**) indicates *p* < 0.001.

**Figure 7 ijms-23-13257-f007:**
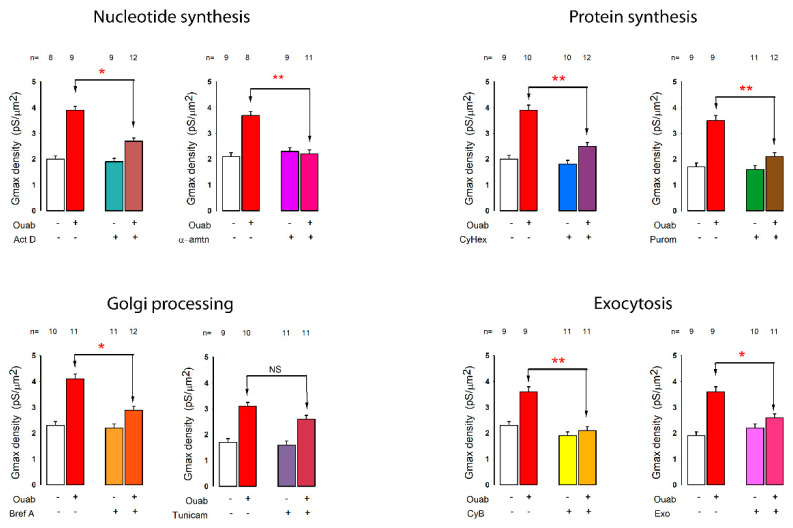
The effect of inhibitors on the synthesis, assembly and export of new proteins involved in the increase in potassium conductance density, caused by ouabain, in cells in mature monolayers. Each bar graph shows the mean (±SE) value of *δ**G_max_*, obtained from current records of individual cells in MEMs, which were treated or not with ouabain for 6 hours and treated or not with the different inhibitors as indicated at the bottom of the bars. The number of repetitions in each group is indicated above each bar. Asterisks indicate a statistically significant difference between the groups indicated by the arrows. (*) indicates a *p* < 0.05 and (**) *p* < 0.001. Abbreviations: Act D = actinomycin D, α-amnt = α-amantidin, CyHex = cycloheximide, Purom = puromycin, Bref A = brefeldine A, Tunicam = tunicamicin, CyB = cytochalasine B. NS: no significance.

**Figure 8 ijms-23-13257-f008:**
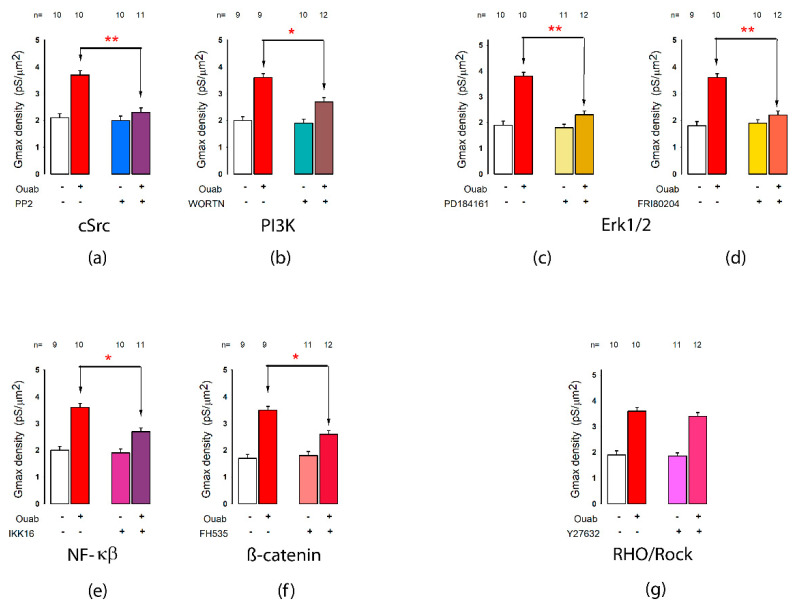
Signaling pathways involved in ouabain-induced changes to I_K_. Bar graphs (**a**–**g**) show the effect on the ouabain-induced enhancement of *δ**G_max_*, of the various kinase inhibitors, as indicated at the bottom of each bar graph. Each bar graph compares the mean (±SE) value of *δ**G_max_*, obtained from current records of individual cells in MEMs which were treated or not with ouabain for 6 hours and treated or not with the different inhibitors as indicated at the bottom of the bars. The number of repetitions in each group is indicated above each bar. Asterisks indicate a statistically significant difference between the groups indicated by the arrows. (*) indicates a *p* < 0.05 and (**) *p* < 0.001.

**Table 1 ijms-23-13257-t001:** Parametric values of potassium currents from cells in MEMs. Values obtained from a sample of 32 cells in MEMs.

Parameter	Mean	SE	Units
*δG_max_*	2.55	0.13	pS/µm^2^
*V* _50_	25.2	1.0	mV
*Hs*	15	0.5	mV^−1^
*τ* _0_	39.9	1.1	ms
*τ_v_*	49.0	3.5	mV

**Table 2 ijms-23-13257-t002:** List of inhibitors used in this work.

Name	Cat. No.	Stock mg/mL	Dose
Actinomycin D	A1410	10	50 nM
α-amanitin	A2263	1	10 μg/mL
Cycloheximide	C7698	25	50 μM
Puromycin	P8833	13	10 μM
Brefeldin A	B7651	10	20 μM
Tunicamycin	T7765	20	10 μg/mL
Cytochalasin B	C6762	20	5 μg/mL
Exo1	E8280	100	50 μM
FR180204	SML0320	25	400 nM
Y-27632	Y0503	30	140 nM
PD 098059	P215	30	7 µM
FH535	F5682	10	15 µM
IKK-16	SML1138	10	200 nM
Wortmannin	W1628	14	10 nM
PP2	P0042	1.4	100 nM

## Data Availability

Not applicable.

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
