# Peer review of "Ouabain-Induced Changes in the Expression of Voltage-Gated Potassium Channels in Epithelial Cells Depend on Cell–Cell Contacts"

_ijms, 2022, doi:10.3390/ijms232113257_

Round 1

Reviewer 1 Report

This is an interesting study; authors should further address the following comments:

 1. Whether glass coverslip treated with cationic solution to make cell adhere?

2. Figure 3: Trypsinization effect between Ouabain and non-treated cells has marginal difference till 12hours of treatment and post 24 hours something is different between two groups. It is not clear why the result interpretation considered as fast recovery of potassium channels?

Author Response

Comments and Suggestions for Authors

This is an interesting study; authors should further address the following comments:

  1. Whether glass coverslip treated with cationic solution to make cell adhere?

ANSWER: No, it is not necessary. For MDCK cells no special treatment is required, only, to sterilize, we bake the pieces of coverslips after cutting.

  1. Figure 3: Trypsinization effect between Ouabain and non-treated cells has marginal difference till 12hours of treatment and post 24 hours something is different between two groups. It is not clear why the result interpretation considered as fast recovery of potassium channels?.

ANSWER:  The interpretation that ouabain promotes faster recovery of channels lost during tripsinization results from (1) In cells treated with ouabain, the maximum conductance density value takes less (12) hours to have a value statistically indistinguishable from the value of cells before tripsinization, while in cells not treated with ouabain 24 hours are required and (2) that the maximum conductance density can Interpreted as an estimate of the density of channels in the membrane. Of course, it is an assumption that requires further work to figure out the molecular entity of the channels involved. Following this comment, I modified Figure 3.B4 to denote the statistical significance of the conductance density values with respect to the value of this variable before tripsinization, in this way the interpretation of the results is more understandable. I also modified the text that describes these results and interpretation.

Reviewer 2 Report

Dear Authors,

I found your manuscript very interesting and well-presented. I have only two minor comments:

Regarding the results on “2.6. Na+/K+-ATPase is the receptor mediating the action of ouabain on potassium currents”. The authors mention that they utilized MDCK-R, a cell line which is rendered unsensitive to ouabain by chemical mutagenesis. After performing their experiments, they come to the conclusion that since MDCK-R cells differ significantly in all of the studied parameters, the Na+/K+-ATPase must be the receptor that mediates the effect of ouabain on potassium currents in MDCK cells. I believe that since these MDCK-R cells are unsensitive to ouabain and not Na+/K+-ATPase-knockouts, the authors should revise their conclusion to something like this “These results indicate that the Na+/K+-ATPase might be the receptor that mediates the effect of ouabain on potassium currents in MDCK cells.” This should be also revised in the Conclusions section.

In addition, I think the paragraph analyzing the result “2.7. Ouabain stimulates the synthesis, assembly, and exportation of new ion channel units.” is not complete, as the authors do not present in the text finding of Figure 7.

Author Response

Comments and Suggestions for Authors

Dear Authors,

I found your manuscript very interesting and well-presented. I have only two minor comments:

Regarding the results on “2.6. Na+/K+-ATPase is the receptor mediating the action of ouabain on potassium currents”. The authors mention that they utilized MDCK-R, a cell line which is rendered unsensitive to ouabain by chemical mutagenesis. After performing their experiments, they come to the conclusion that since MDCK-R cells differ significantly in all of the studied parameters, the Na+/K+-ATPase must be the receptor that mediates the effect of ouabain on potassium currents in MDCK cells. I believe that since these MDCK-R cells are unsensitive to ouabain and not Na+/K+-ATPase-knockouts, the authors should revise their conclusion to something like this “These results indicate that the Na+/K+-ATPase might be the receptor that mediates the effect of ouabain on potassium currents in MDCK cells.” This should be also revised in the Conclusions section.

ANSWER: Thank you very much for your comment, it is true that the results, although they suggest, are not indisputable evidence, so their interpretation should be considered a suggestion rather than a sentence. Accordingly, I have rephrased the text, both in the abstract, in the results section and in the conclusions section.

In addition, I think the paragraph analyzing the result “2.7. Ouabain stimulates the synthesis, assembly, and exportation of new ion channel units.” is not complete, as the authors do not present in the text finding of Figure 7.

ANSWER: It is true, due to an unfortunate mistake there was a part of the text of that section that was not added during the preparation process. Thank you very much for your wise remark. I have already added the missing text to the manuscript